# Antibacterial Activity and Amphidinol Profiling of the Marine Dinoflagellate *Amphidinium carterae* (Subclade III)

**DOI:** 10.3390/ijms222212196

**Published:** 2021-11-11

**Authors:** Maria Elena Barone, Elliot Murphy, Rachel Parkes, Gerard T. A. Fleming, Floriana Campanile, Olivier P. Thomas, Nicolas Touzet

**Affiliations:** 1Centre for Environmental Research, Sustainability and Innovation, Department of Environmental Science, School of Science, Institute of Technology Sligo, Ash Ln, Ballytivnan, F91 YW50 Sligo, Ireland; MariaElena.Barone@mail.itsligo.ie (M.E.B.); Rachel.Parkes@mail.itsligo.ie (R.P.); 2Marine Biodiversity, School of Chemistry, Ryan Institute, National University of Ireland Galway (NUI Galway), University Road, H91 TK33 Galway, Ireland; E.MURPHY53@nuigalway.ie; 3Discipline of Microbiology, School of Natural Science, National University of Ireland Galway (NUI Galway), University Road, H91 TK33 Galway, Ireland; ger.fleming@nuigalway.ie; 4Medical Molecular Microbiology and Antibiotic Resistance Laboratory (MMARLab), Department of Biomedical and Biotechnological Sciences, University of Catania, Via Santa Sofia n. 97, 95123 Catania, Italy; F.campanile@unict.it

**Keywords:** microalgae, *Amphidinium carterae*, antimicrobial activity, marine natural products (MNP), minimum inhibitory concentration (MIC), minimum bactericidal concentration (MBC), amphidinols, UHPLC-HRMS, targeted metabolomics

## Abstract

Microalgae have received growing interest for their capacity to produce bioactive metabolites. This study aimed at characterising the antimicrobial potential of the marine dinoflagellate *Amphidinium carterae* strain LACW11, isolated from the west of Ireland. Amphidinolides have been identified as cytotoxic polyoxygenated polyketides produced by several *Amphidinium* species. Phylogenetic inference assigned our strain to *Amphidinium carterae* subclade III, along with isolates interspersed in different geographic regions. A two-stage extraction and fractionation process of the biomass was carried out. Extracts obtained after stage-1 were tested for bioactivity against bacterial ATCC strains of *Staphylococcus aureus*, *Enterococcus faecalis*, *Escherichia coli* and *Pseudomonas aeruginosa*. The stage-2 solid phase extraction provided 16 fractions, which were tested against *S. aureus* and *E. faecalis.* Fractions I, J and K yielded minimum inhibitory concentrations between 16 μg/mL and 256 μg/mL for both Gram-positive. A targeted metabolomic approach using UHPLC-HRMS/MS analysis applied on fractions G to J evidenced the presence of amphidinol type compounds AM-A, AM-B, AM-22 and a new derivative dehydroAM-A, with characteristic masses of *m/z* 1361, 1463, 1667 and 1343, respectively. Combining the results of the biological assays with the targeted metabolomic approach, we could conclude that AM-A and the new derivative dehydroAM-A are responsible for the detected antimicrobial bioactivity.

## 1. Introduction

The emergence of antimicrobial resistance has hindered the effectiveness of treatments for a growing number of bacterial infections worldwide [1,2]. The overuse and misuse of antibiotic drugs have led bacteria to develop adaptations to overcome the mechanisms of action of several commonly used drugs [3,4], hence requiring the urgent identification of novel antimicrobial compounds. Several bioactive metabolites isolated from marine bioresources, also known as marine natural products (MNPs), have elicited potent bioactivity against cancer and other ailments induced by pathogens such as viruses, bacteria and fungi [5,6,7]. Antimicrobial MNPs show a high chemical diversity and include, for example, alkaloids, terpenoids, peptides, halogenated compounds, polyketides, isocoumarins, or nucleosides [6,8].

Microalgae are photosynthetic organisms increasingly exploited in the context of bioenergy, bioremediation or the biorefinery of high-value compounds [9]. Some species can biosynthesise and accumulate compounds such as polysaccharides, pigments, proteins, vitamins, polyunsaturated fatty acids, antioxidants, and other bioactive molecules [9,10]. Microalgae have also been increasingly screened for new antibacterial drugs [6,11,12,13,14,15,16]. Compounds such as cyanovirin, oleic acid, linoleic acid, palmitoleic acid, β-carotene, fucoxanthin or phycocyanin do exhibit antioxidant or anti-inflammatory properties as well as antimicrobial activity, for example, against *Staphylococcus aureus* and Methicillin-Resistant *Staphylococcus aureus* (MRSA) [14,17,18,19].

Several marine dinoflagellates are known producers of potent bioactive compounds and some of them are biosynthesise biotoxins that can render shellfish unsafe for human consumption [20], including polyketides, ladder-shape polyethers, spirolides or alkaloids [21,22]. Polyhydroxylated polyketides, named amphidinols, have been identified in species of the marine dinoflagellate genus *Amphidinium* and have previously shown anti-fungal, anti-tumour and anti-bacterial activity [13,23,24,25,26,27,28]. In this context, this study aimed at assessing the antibacterial potential of a strain of the dinoflagellate *Amphidinium*
*carterae* isolated from the west of Ireland. The fractions obtained sequentially using solvents of varying polarity in solid phase extraction were tested against several bacterial pathogens. Due to the low amount of material available, the main amphidinols present in the bioactive fractions were targeted and identified by a targeted metabolomic analysis using UHPLC-HRMS/MS.

## 2. Results

### 2.1. Phylogenetic Characterisation of Strain A. carterae LACW11

Sequencing of the D1D2 domain of the LSU rRNA gene and subsequent BLAST analysis indicated strain LACW11 to belong to the species *Amphidinium carterae*. The phylogenetic tree showed this strain to group with other members of the species in a sister clade to that including *A. trulla*, *A. gibbosum*, *A. massarti*, *A. tomasi*, *A. theodori* and *A. thermaeum* (Figure 1). The four subgroups of *Amphidinium carterae* characterised in previous studies were visible; the Irish strain LACW11 clustering with other isolates from Europe, Canada, Brazil, Puerto Rico and Australia within subclade III.

### 2.2. Amphidinium carterae Culture

The culture of *A.*
*carterae* LACW11 was maintained for 30 days in f/2 medium and harvested during the exponential phase (Figure 2). The recovered freeze-dried biomass was estimated at 850 mg for 6.1 L of culture.

### 2.3. Stage-1: Bioactivity Assays on the Extracts

The freeze-dried biomass of *Amphidinium carterae* LACW11 was first extracted with Et_2_O then MeOH. The methanol extract was then partitioned between EtOAc and H_2_O to provide three final extracts E_Et2O_, E_EtOAc_ and E_H2O_. These extracts were tested for antimicrobial activity (Figure 3).

The assays for stage-1 samples showed moderate antimicrobial bioactivity for extracts E_Et2O_ and E_EtOAc_ against *Staphylococcus aureus* and *Enterococcus faecalis*, and low activities for the E_H2O_ extract. For E_EtOAc_, the MIC values against *S. aureus* and *E. faecalis* were 256 and 512 μg/mL, respectively. Only E_EtOAc_ returned promising MBC values of 512 and 1024 μg/mL against *S. aureus* and *E. faecalis*, respectively. E_H2O_ showed minor antimicrobial bioactivity against the Gram-negative bacterium *E. coli*, with a MIC of 8192 μg/mL. No bioactivity was detected against *P. aeruginosa*.

### 2.4. Stage-2: Bioactivity Assays on the Fractions

#### 2.4.1. Microbial Assays

Stage-2 focused on the most bioactive extract E_EtOAc_, which was further fractionated via C18 solid phase extraction into 16 fractions using mixtures of solvents of decreasing polarities (H_2_O:MeOH:EtOAc). The resulting fractions were tested against the two Gram positive bacteria (Figure 4).

The four fractions H, I, J and K, demonstrated some anti-bacterial activity against *S. aureus* and *E. faecalis* (MICs from 16 to 256 μg/mL). Fractions I and J showed good bactericidal activity against *S. aureus* with MBC value of 32 μg/mL. In comparison, *E. faecalis* showed lower susceptibility to fractions I and J, which were still bacteriostatic, returning higher MIC and MBC values than those obtained for *S. aureus*.

#### 2.4.2. Chemical Profiling of the Bioactive Fractions

Fractions G to K were first analysed by HPLC-DAD-ELSD, evidencing major compounds by ELSD with UV profiles at maximum wavelength 210 nm (Appendix A). Due to the low amount of material available, a purification process could not be envisioned, and we used previously described *m/z* data and MS/MS fragmentation patterns in this family to assess the chemical composition of these five fractions. The UV profiles were then used as comparative data in UHPLC-DAD-HRMS/MS to obtain the MS spectra of these major compounds. Amphidinol type compounds were detected as major compounds in the bioactive fractions G–J due to characteristic *m/z* and fragmentation patterns, but these compounds were absent in fraction K [29]. The retention times for the compounds of interest ranged from 3.4–4.1 min with *m/z* at 1667, 1361, 1463 and 1343 during a 16 min UHPLC run (Table 1). The main fragments observed in the MS/MS fragmentation spectra of the compounds at *m/z* 1343, 1361 and 1463 [M + Na]^+^ were found at *m/z* 1085 and 687, while the main fragments of amphidinols/luteophanols were shown to be at *m/z* 903 or 1105 following the numbering recently proposed by Wellkamp et al. [29]. The only known derivatives with fragments at *m/z* 1085 (C-29/C-30) and 687 (C-29/C-30 and C-1/C-1′) were identified as amphidinol A (*m/z* 1361) and B (*m/z* 1463) from the comprehensive analysis of Wellkamp et al. [29] and Cutigano et al. [28]. A comparison between the fragmentation patterns of these compounds confirmed their identity. A third derivative at *m/z* 1668 [M + Na]^+^ could be assigned to amphidinol-22, the only derivative with this molecular mass. The fragmentation pattern was also very similar to this known compound according to Martinez et al. [25].

The collision induced MS/MS dissociation spectra obtained from fractions G, H, I and J evidenced the presence of some previously reported amphidinols [29]. Three already known compounds, AM-A, AM-B and AM-22, were found in fractions G, H and I. Another major derivative with *m/z* 1343.8447 was found in fraction J. The fragmentation pattern of this compound was very similar to an unknown derivative, named N16, in Wellkamp et al. [29] with a main fragment at *m/z* 1085 and no other fragment at about 1100.

The mass of this compound indicated that it could correspond to a dehydro derivative of AM-A. The characteristic fragmentation pattern of AM-A leads to two major fragments at *m/z* 1143 (C-32/C-33 fragmentation) and 1085 that follow a McLafferty rearrangement around the ketone at C-31. The change in fragmentation pattern between the new derivative dehydroAM-A (N16) and AM-A with an absence of the fragment at *m/z* 1143 suggests that the dehydration might occur at position C-33, but NMR data are needed to confirm this assumption.

A comparative study was then performed on the different collected fractions targeting only the main amphidinol derivatives. A relative integration of the amphidinol derivatives was measured in the different fractions, considering the similar ionization potential between these analogues (Table 2). AM-B appeared as the major amphidinol in fractions G and H. AM-A was found as the major amphidinol derivative in fraction I and dehydroAM-A in fraction J. As both fractions are the most active on *S. aureus,* we can conclude that both compounds are active on this strain. Even though fraction K was found to be active on the two Gram-positive strains, we could not detect any amphidinol derivative. This activity should therefore originate from other metabolites to be further investigated.

## 3. Discussion

There has been a pressing demand worldwide for the identification of new bioactive compounds to address the emerging issue of antibiotic resistance, which is associated with an increasing number of bacterial pathogens in both veterinary and human health settings [4,30,31]. The bioprospecting and screening of extracts obtained from natural sources, including marine organisms such as bacteria, microalgae, sponges, or molluscs, have led to the identification of pharmacologically active compounds with the potential for translation into novel drugs [5,13,21,24,25]. In particular, recent studies have reported the identification of various new bioactive from marine microbial sources [5,13,32].

Microalgae constitute a polyphyletic and heterogeneous group of protists that have been portrayed as promising bioresources to further exploit owing to the scalability of their cultivation and the range of bioactive molecules they are known to biosynthesise [9,33]. The marine dinoflagellate genus *Amphidinium* has been of interest for the production of several bioactive compounds [34,35,36].

*Amphidinium* species such as *A. carterae*, *A. massartii*, *A. klebsii* and *A. operculatum* also produce molecules with antimicrobial potential, such as polyhydroxylated polyketides, which encompass an array of chemically related compounds such as amphidinols, amphidinolactones, lasonolide, iriomoteolides, amphirions, colopsinols, amphezonol, luteophanols or karatungiols [5,21,22,23,24,25,37,38,39,40,41,42]. There is a degree of genetic diversity among *A. carterae* isolates that has been previously revealed via the analysis of the LSU rRNA gene, leading to the delineation of four subgroups based on the clustering of specific strains within defined clades in phylogenetic inferences [32,43]. Our strain, *A. carterae* strain LACW11 from the west of Ireland, grouped with other strains of subclade III, including strain DN241EHU previously analysed by Wellkamp et al. [29]. These strains seem to be the only isolates originating from the northwest European Atlantic area so far. Other species complexes have previously been identified in marine dinoflagellates, such as the Harmful Algal Bloom (HAB) species *Alexandrium tamarense*, and sub-groups tend to show distinct biogeographic patterns in their global distributions [44]. There are far less sequences of *Amphidinium* available in GenBank compared to HAB species, but the isolates of *A. carterae* that cluster together with strain LACW11 within subclade-III appear to be globally distributed. In the context of potential inter-strain variability in the bioactivity of extracts of distinct isolates of this species, and of their chemical make-up, akin to the existing variability of toxin profiles among some HAB species [45,46], questions do arise with regard to their composition in amphidinols, which could potentially be used as chemical markers for the four previously identified subclades.

Amphidinols were first described by Satake et al. [26] from purified extracts of the species *Amphidinium klebsii*. While recent studies have focused on the profiling of a different family of polyketide macrolides named amphidinolides, also produced in certain strains of *Amphidinium* and exhibiting high cytotoxicity, there are fewer reports on the polyhydroxylated polyketides of the amphidinol family, especially on their antimicrobial potential. Of significance, Morales-Amador et al. [47] were able to purify AMs 20B, 24, 25, 26 and luteophanol D from the medium of a batch-culture of the strain *A. carterae* ACRN03, isolated from the Indian Ocean in La Reunion Island. Martinez et al. [25] on the other hand worked with the strain of American origin, CCMP1314, and confirmed the presence of AMs 18, 19 and 22. The latter was purified, and subsequent bioactivity tests showed that these compounds do not exhibit antibacterial activity against MRSA, an MSSA strain of *S. aureus*. In our work, the collision induced dissociation spectra (MS/MS) obtained from fractions G-J from *A. carterae* LACW11 evidenced the presence of previously identified and reported amphidinols AM-A, AM-B and AM-22. Using a comparison of fragmentation patterns with known amphidinols, we suggest that a dehydroAM-A derivative, also characterized as N16 by Wellkamp et al. [29], would be the major component of fraction J. Fractions I and J exhibited the highest antibacterial activity against *S. aureus* and *E. faecalis*. As shown by the ELSD profiles, the major amphidinols present in our strain were AM-A (Fraction I) and DehydroAM-A (Fraction J). Interestingly, this amphidinol profile shows a very high level of similarity with that of strain DN241EHU collected in Mallorca (Spain) in the Mediterranean Sea, as reported by Wellkamp et al. [29]. This strain is the only one containing the dehydro derivative N16 of amphidinol A and therefore the Irish strain represents the second example of a strain producing this derivative. The combination of chemical profiles and the bioactivities of the fractions leads to the conclusion that AM-A and its new dehydro derivative could be responsible for the antibacterial activity detected in the extracts. These results also confirm that AM-22 does not exhibit a significant antibacterial activity on the tested strains, but also that non-amphidinol derivatives should be responsible for antibacterial activity in fraction K.

Previous studies reported the isolation and structure elucidation of metabolites from *Amphidinium* sp. with bactericidal, fungicidal, anti-cancer and hemolytic bioactivities [21,24,25,48]. Here, the initial E_EtOAc_ extract of *A. carterae* LACW11 showed promising bioactivity against Gram-positive bacteria, in particular *S. aureus*. Then, the two fractions I and J resulting from the SPE-based purification process, showed inhibitory and bactericidal activities against *S. aureus* at 16 µg/mL and 32 µg/mL, respectively. Kubota et al. [49] reported similar MICs, albeit using purified compounds, for Amphidinins C–F and Amphidinolide Q extracted from *Amphidinium* sp. (2012-7-4A strain) against *Escherichia coli*, *Staphylococcus aureus*, *Bacillus subtilis*, *Aspergillus niger*, *Trichophyton mentagrophytes* and *Candida albicans*. Other polyketide derivatives extracted from *Amphidinium* species, such as karatungiols or amphirionin-2, have also shown bioactivity against fungi and human cancer cell lines [41,50]. Interestingly, amphidinol-3 showed a mechanism of action similar to that of other polyene antibiotics such as amphotericin B and filipin [27]. It has been indicated that amphidinols may interact with membranes, leading to their permeabilization via pore formation and subsequent cell death [51,52]. It is noteworthy to mention that our amphidinol-containing extracts showed much greater activity against *S. aureus* and *E. faecalis* compared to *E. coli* and *P. aeruginosa*, which may reflect the fundamental differences in cell membrane and peptidoglycan characteristics between Gram-positive and Gram-negative bacteria. Antimicrobial activity against *E. faecalis* was also observed in the present study, with MIC and MBC values ranging from 64 to 256 µg/mL and 1024 to 8192 µg/mL, respectively, showing bacteriostatic activity similar to that of some macrolides (e.g., erythromycin), which can become bactericidal at higher doses [53,54]. It is noteworthy that some bioactivity against *E. faecalis* was recorded for fraction K, in which no amphidinol was detected. Moreover, fractions G and H, which both contained amphidinol-B, returned very moderate antibacterial activities. These results indicate the presence of other bioactives in these fractions, and/or potential synergistic effects with other compounds. Future work will focus on purifying the new dehydroAM-A detected in our strain and on characterizing both its structure and antibacterial activity. Subjecting the microalgal cells to varying incubation regimes may also lead to the potential observation of variations in the profile of the amphidinols this species can synthesize.

## 4. Materials and Methods

### 4.1. Amphidinium carterae Cultivation

A batch culture of *Amphidinium carterae* LACW11 isolated from the northwest of Ireland was prepared in a sterile 10 L glass bottle (6.1 L final volume culture) fitted with a 2-port vented cap at a starting concentration of 0.5 mg/mL (wet biomass) in f/2 medium without silicate [55,56]. The culture was incubated for 30 days at 20 ± 1 °C under ca. 60–80 μmol/m^2^/s illumination provided by LED panels (white light) and a 14:10 light:dark photoperiod. Aeration through a 0.22 µm filter airline was provided at a rate of 210 mL/min. On day 30, the cells were harvested by centrifugation at 2000 rpm for 5 min to collect the biomass, which was then desalted with 1 mL of 0.5 M ammonium formate prior to overnight freeze-drying (Scanvac. MillRock, Kingston, NY, USA) and subsequent storage at −20 °C.

### 4.2. DNA Extraction, Partial 28SrDNA Gene PCR and Sequencing

DNA extraction was carried out using the E.Z.N.A. ^®^Plant DNA kit (Omega Bio-Tek, Norcross, GA, USA). PCR targeting the D1D2 domain of the 28S rRNA ribosomal gene was performed using the 1X DreamTaq™ Green PCR Master MIX (Thermo Fischer Scientifics, Baltics, Vilnius, Lithuania) using the primers D1R (forward, 5′ ACCCGCTGAATTTAAGCATA 3′) and D2C (reverse, 5′ CCTTGGTCCGTGTTTCAAGA 3′) [57]. The thermocycling program was as follows: 94 °C for 3 min then 35 cycles consisting each of 94 °C for 1 min (denaturation), 52 °C for 1 min (annealing) and 72 °C for 3 min (extension). A final extension step of 72 °C for 6 min was included prior to gel electrophoresis that was carried out using a 1% agarose 1X TEA buffer gel stained with Gel Red and observed under UV light in a transilluminator. The amplicon was purified using the E.Z.N.A^®^ Cycle Pure Kit (Omega BIO-TEK, Norcross, GA, USA) prior to external sequencing (MWG-Eurofins, Koln, Germany).

### 4.3. Phylogenetic Inference

The *Amphidinium carterae* LACW11 sequence was run through BLAST against other deposits in the NCBI database. Several *Amphidinium* sp. and dinoflagellate LSU rDNA sequences were imported from GenBank to generate a multiple sequence alignment using Clustal W and Mega X [58]. A phylogenetic tree was constructed using a Neighbor-Joining matrix with a Tamura-Nei model and a discrete Gamma distribution (TN93 + G) [59]. Maximum likelihood was chosen using a number of three threads on nucleotide substitution. The robustness of the tre1000 replicates.

### 4.4. Extraction and Fractionation of the Biomass

The extraction and antimicrobial activity tests were carried out in two phases. For stage 1, 850 mg of freeze-dried biomass was homogenised for 2 min with 55 mL of diethyl ether (Et_2_O), then incubated at 4 °C for 24 h. The extract was centrifuged at 2000 rpm for 3 min and dried using a Rotavap (E_Et2O_). The residual cell materials collected after centrifugation of the initial Et_2_O extract were further extracted in 55 mL of methanol (MeOH) for 24 h at 4 °C. Following centrifugation at 2000 rpm for 3 min, the corresponding extract was dried under reduced pressure. A liquid–liquid partition (1:1 *v/v*) was then carried out on this residue using 2 mL of ethyl acetate (EtOAc) and 2 mL of deionised water (H_2_O), returning two phases which were separately collected as fractions E_EtOAc_ and E_H2O_, respectively. All final three extracts were dried and weighed prior to storage at −20 °C and subsequent antimicrobial activity tests.

Stage 2 focused on extract E_EtOAc_ based on the result of the first round of antimicrobial tests. E_EtOAc_ (108 mg) was resuspended in EtOAc and transferred to a 50 mL round bottomed flask containing 600 mg of C18 powder (POLYGOPREP 60–50 C_18_). The suspension was dried under reduced pressure. A C_18_ SPE Cartridge (Agilent Bond Elut Mega BE-C18 1 g 6 mL) was conditioned with MeOH (2 × 6 mL) followed by H_2_O (2 × 6 mL). Once conditioned, the dried sample was loaded to the column. In total, 16 fractions (12 mL each) were recovered by eluting solvents of decreasing polarity: (A) 100% H_2_O, (B) H_2_O:MeOH (90:10 *v/v*), (C) H_2_O:MeOH (80:20 *v/v*), (D) H_2_O:MeOH (70:30 *v/v*), (E) H_2_O:MeOH (60:40 *v/v*), (F) H_2_O:MeOH (50:50 *v/v*), (G) H_2_O:MeOH (40:60 *v/v*), (H) H_2_O:MeOH (30:70 *v/v*), (I) H_2_O:MeOH (20:80 *v/v*), (J) H_2_O:MeOH (10:90 *v/v*), (K) 100% MeOH, (L) MeOH:EtOAc (80:20 *v/v*), (M) MeOH:EtOAc (60:40 *v/v*), (N) MeOH:EtOAc (40:60 *v/v*), (O) MeOH:EtOAc (20:80 *v/v*) and (P) 100% EtOAc. Samples were dried under reduced pressure and weighed prior to storage for further chemical and antibacterial activity analyses.

### 4.5. Chemical Profiling of the Fractions by LC-MS

Approximately 0.2 mg of dried extract of the 16 fractions (A to P) was resuspended in the corresponding fraction solvent at a concentration of 1 mg/mL prior to chemical profiling.

Chemical profiling was first performed on a HPLC-DAD-ELSD (Agilent Infinity 1260 Quat. pump and UV-DAD, Agilent technologies 385-ELSD). The column used was a 4.6 mm × 250 mm i.d., 5 µm, symmetry C_18_ (Waters, Wexford, Ireland). The gradient was from 30% B until 5 min, 30–90% B over 20 min and held for 7 min, then returned to 30% B over 2 min at 1.0 mL/min, held for 1 min, and returned to the initial conditions over 1 min and held for 5 min to equilibrate the system. UV detection was performed at λ 210, 254 and 290 nm. The injection volume was 30 µL and the column and sample temperatures were set at 40 °C and 10 °C, respectively.

Samples were then run in positive MS^e^ (200–3000 *m/z*) and MS/MS modes. High resolution mass spectra data were obtained with an Agilent 6540 qTof mass spectrometer UHPLC-DAD-HRMS. MS/MS data used a cone voltage was 40 V, collision energy was 75 V. The cone and desolvation gas flows were set at 300 and 12 L/min, respectively, and the source temperature was 300 °C. A binary gradient elution was used, with phase A consisting of water and phase B of acetonitrile in water (both containing 6.7 mM ammonium formate). The column used was a 50 mm × 2.1 mm i.d., 1.7 µm, Acquity UPLC BEH C_18_ (Waters, Wexford, Ireland). The gradient was from 30–90% B over 11 min at 0.4 mL/min, held for 1 min, and returned to the initial conditions over 1 min and held for 2 min to equilibrate the system. The injection volume was 5 µL and the column and sample temperatures were 40 °C and 10 °C, respectively. HRMS spectra were obtained using the LCMS conditions described by Welkamp M. et al. [29]. All fractions were analysed by UHPLC-HRMS/MS.

### 4.6. Microbiological Assays

#### 4.6.1. Bacterial Strains

Microbiological assays were carried out using the Gram-positive *S. aureus* ATCC 25,923 and *E. faecalis* ATCC 29,212, and the Gram-negative *E. coli* ATCC 25,922 and *P. aeruginosa* ATCC bacterial controls. The strains were grown on solid agar media then incubated at 37 °C overnight prior to carrying out the bioassays. The strains were grown in selective agar media: Bile Aescuilin Azide (BEA) for *E. faecalis*, Mannitol Salt Agar (MSA) for *S. aureus* and MacConkey agar for both *E. coli* and *P. aeruginosa*. All media were manufactured by Oxoid (Basingstoke, UK).

#### 4.6.2. Determination of Minimum Inhibitory Concentrations (MIC)

The susceptibility tests were carried out using the broth microdilution assay based on the CLSI 2020 guidelines [60]. One colony of each ATCC strain was inoculated in 5 mL of BHI (Brain Heart Infusion broth) for 4 h prior to carrying out a dilution at a final concentration of 10^5^ cfu/mL for inoculation of the wells of a 96-well plate in 0.1 mL of sterile Muller-Hinton broth (MHB).

For the stage 1 fractions, the dried microalgal extracts were re-suspended in 4% DMSO for fraction E_H2O_, and 10% acetone for fractions E_Et2O_ and E_EtOAc_ (*n* = 3).

For the stage 2 fractions, all the extracts (A to P) were re-suspended in 10% acetone. Prior to bacterial inoculation, scalar dilutions (1:2) of the extracts were performed from the 1st to the 11th wells of a row of a 96-well plate with a final volume of 100 μL of MHB. The 12th well was used as a bacterial growth control.

The antibiotics, kanamycin and colistin, were used as positive controls for the Gram-positive and negative bacterial species, respectively. Control wells of MHB and solvents were also included. Triplicate wells were used for all the samples and the plates were incubated at 37 °C for 14–18 h. Nitro blue-tetrazolium (NBT) (1 mg/mL) was then added to each well and incubated for 30 min to determining the MIC points, based on the ability of bacteria to reduce NBT to formazan, as previously described [61].

#### 4.6.3. Determination of Minimum Bactericidal Concentration (MBC)

Wells above the MIC value were plated in Muller-Hinton agar and incubated at 37 °C overnight. The MBC value was identified by determining the lowest concentration of antibacterial agent that kills ≥ 99.9% of the bacterial population. Antibacterial agents are usually regarded as bactericidal if the MBC value is no more than 4-fold the MIC value.

## 5. Conclusions

SPE-based fractionation of extracts from an Irish isolate of *Amphidinium carterae*, which grouped with strains from other geographic areas within sub-clade III, returned bioactivity against the Gram-positive bacteria *S. aureus* and *E. faecalis*. The activity against *S. aureus* was mostly associated with fractions I and J, where amphidinols AM-A and dehydroAM-A were prominent. Fraction K returned minor activity against *E. faecalis,* but did not contain known amphidinols, while fractions G and H, which contained AM-B, also did not return noteworthy antibacterial activity. Further fraction purification and testing of individual compounds is warranted.

## Figures and Tables

**Figure 1 ijms-22-12196-f001:**
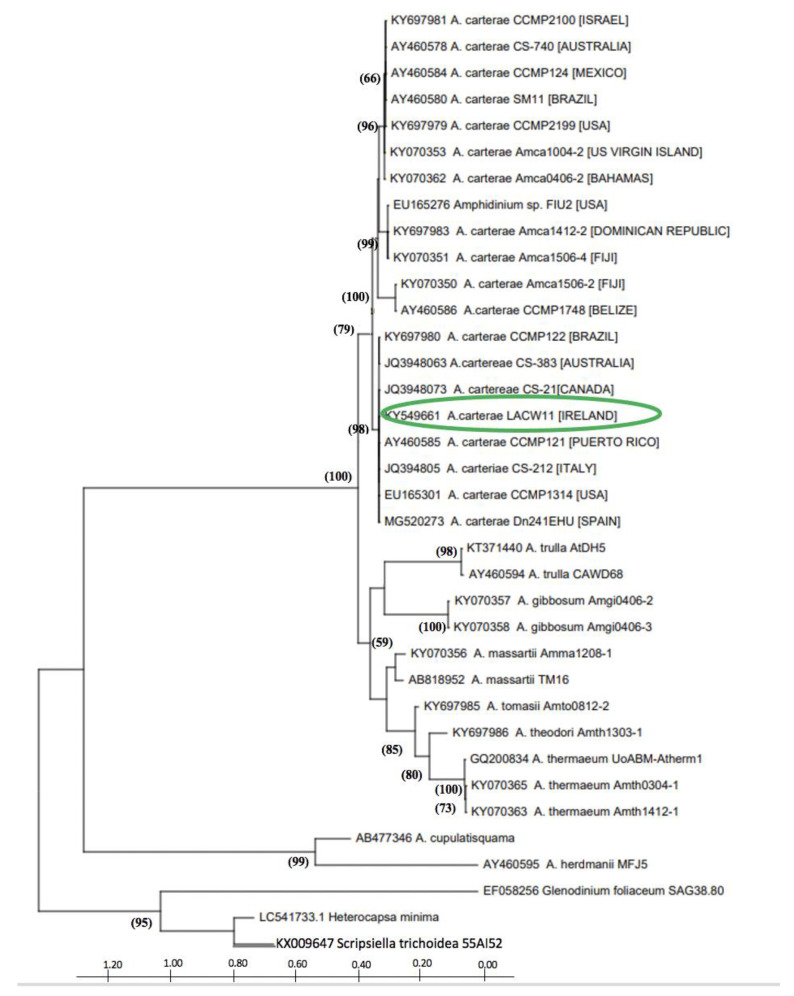
Phylogenetic tree inferred from the maximum likelihood analysis of partial LSU rDNA sequences of *Amphidinium* species. The optimal base substitution model derived from the Bayesian Information Criterion (BIC) using MEGA X was Tamure Nei gamma distributed (TN93 + G, parameter = 0.4032). Bootstrap values (%) of 1000 replicates are shown (only values > 50). The position of strain LACW42 in the tree is highlighted in green.

**Figure 2 ijms-22-12196-f002:**
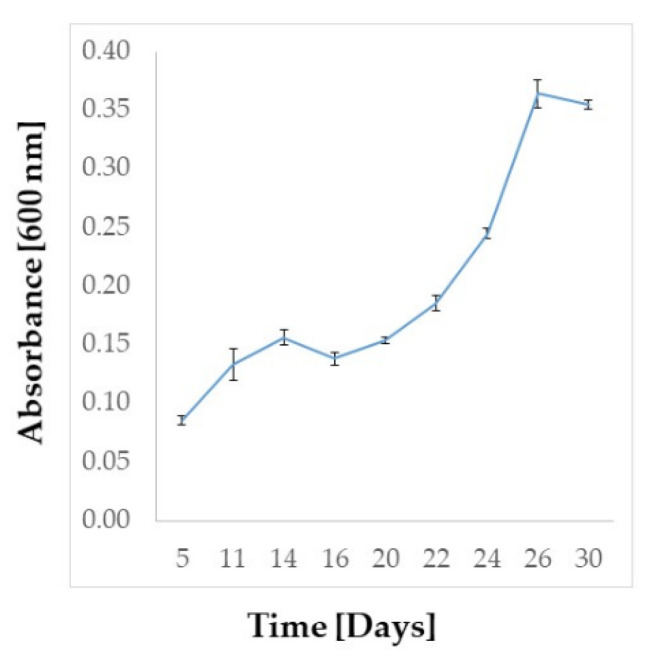
Batch growth of *Amphidinium carterae* LACW11 in f/2 medium monitored at λ = 600 nm.

**Figure 3 ijms-22-12196-f003:**
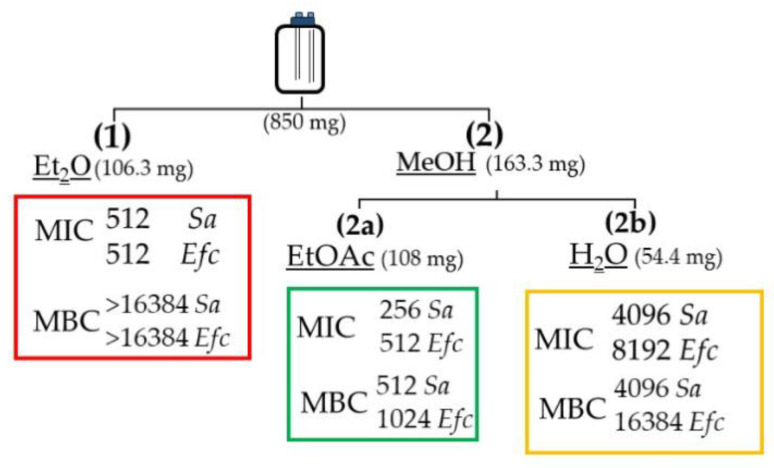
Representative scheme of Stage-1: three extracts obtained from *Amphidinium carterae* LACW11 biomass (freeze dried, mg) were screened against Gram-positive bacteria. Activities are expressed in µg/mL. *Sa*, *Staphylococcus aureus* ATCC 25,293; *Efc*, *Enterococcus faecalis* ATCC 29,212. The bioactivities of each extract are highlighted with different colours.

**Figure 4 ijms-22-12196-f004:**
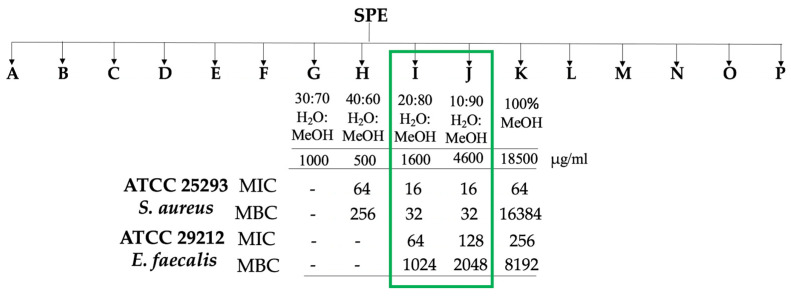
Representative scheme of Stage-2: fractions obtained via C18 Solid Phase Extraction (SPE) of the E_EtOAc_ from *Amphidinium carterae* LACW11, which were tested against Gram-positive bacteria. Activities are expressed in µg/mL. The bioactivities of fractions I and J are highlighted.

**Table 1 ijms-22-12196-t001:**
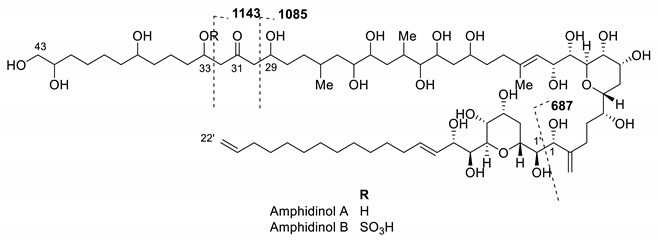
Amphidinol derivatives present in fractions G–J.

Compound	Fraction	*m/z* Observed	Elemental Composition	RT (min)	±ppm
AM-B	G, H	1463.7927	C_69_H_125_Na_2_O_27_S	4.125	0.192
AM-22	I	1667.9270	C_84_H_140_NaO_31_	3.473	0.705
AM-A	I, J	1361.8547	C_69_H_126_NaO_24_	3.649	1.395
Dehydro-AM-A	J	1343.8447	C_71_H_122_O_23_	3.766	1.015

**Table 2 ijms-22-12196-t002:** Relative proportions of the main amphidinol derivatives identified in fraction G–K. MIC values are expressed in µg/mL.

	Stage-2 Fractions				
	G	H	I	J	K
MIC *S. aureus*	-	64	16	16	64
MIC *E. faecalis*	-	-	64	128	256
AM-B	100	100	0	0	0
AM-22	0	0	8	0	0
AM-A	0	0	92	8	0
DehydroAM-A	0	0	0	92	0

## Data Availability

Not applicable.

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
