# Peer review of "Antibacterial Activity and Amphidinol Profiling of the Marine Dinoflagellate Amphidinium carterae (Subclade III)"

_ijms, 2021, doi:10.3390/ijms222212196_

Round 1
Reviewer 1 Report
This paper describes the production of amphidinol, antibacterial chemicals, by a dinoflagellate Amphidinium carterae, and shows some of them had very strong activities against Staphylococcus aureus and Enterococcus faecalis. Data are clear and conclusion is also appropriate.
A reviewer supposes this article is acceptable after some minor corrections.
- Why these amphidinols inhibit only the Gram-positive bacteria and no effects on the Gram-negative bacteria? A reviewer suggests this point should be commented in the Discussion.
- Production of such bioactive compounds by microalgae often depend on the culturing condition and the algal growth phase. The authors described culturing conditions (medium, temperature, and illumination) but no details on the algal growth stage. What growth phase is “on day 30”? Please add more description on the method of preculturing and others in Materials and Methods.
- After getting such successful results as described in the article, then what do you intend to continue the study? Please add some future plan to expand this study at the last part of the Discussion.
- All the scientific names of algae and bacteria in Figs. 2 and 3 should be written in Italic in the figure legends.
Author Response
Dear Reviewer,
Many thanks for reviewing our manuscript entitled “Antibacterial activity and amphidinol profiling of the marine dinoflagellate Amphidinium carterae (subclade III) by Maria Elena Barone – Elliot Murphy, Rachel Parkes, Gerard T.A. Fleming, Floriana Campanile, Olivier P. Thomas and Nicolas Touzet (ijms-1440142).
Please find below our responses to the comments. We hope that the queries have been addressed satisfactorily.
Please do not hesitate to get back to us should further clarifications be needed.
On behalf on the authors,
Yours faithfully,
Maria Elena Barone
#1 Why these amphidinols inhibit only the Gram-positive bacteria and no effects on the Gram-negative bacteria? A reviewer suggests this point should be commented in the Discussion.
Response: This is a good point indeed. There is not much yet in the literature regarding the antibacterial activity of amphidinols, even less so with respect to their specific mode of action. Other studies have indicated that amphidinols may interact with membranes, leading to their permeabilization via pore formation and cell death. Furthermore, this suggest the possibility that the resistance of Gram-negative bacteria could be due the inability of amphidinols to permeate the OM and/or to their inability to interact with the cytoplasmic membrane.
This point is now briefly emphasized in the discussion (l-257).
#2 Production of such bioactive compounds by microalgae often depend on the culturing condition and the algal growth phase. The authors described culturing conditions (medium, temperature, and illumination) but no details on the algal growth stage. What growth phase is “on day 30”? Please add more description on the method of preculturing and others in Materials and Methods.
Response: the growth curve was added to the main body of the manuscript together with some brief information in section 2.2.
#3 After getting such successful results as described in the article, then what do you intend to continue the study? Please add some future plan to expand this study at the last part of the Discussion.
Response: our plan of action for further studies was initially placed in the conclusion section. This is now also mentioned as requested at the end of the discussion (l-269).
#4 All the scientific names of algae and bacteria in Figs. 2 and 3 should be written in Italic in the figure legends.
Response: the format of the figure captions was amended as requested.

Reviewer 2 Report
International Journal of Molecular Sciences (Manuscript ID: ijms-1440142), Comments to the Authors:
Title: Antibacterial activity and amphidinol profiling of the marine dinoflagellate Amphidinium carterae (subclade III)
Comments:
The submitted review discussed the identification of amphidinolides. Phylogenetic inference assigned the strain to Amphidinium carterae subclade III along with isolates interspersed in different geographic regions. A two-stage extraction and fractionation process of the biomass was carried out. Extracts obtained after stage-1 were tested for bioactivity against bacterial ATCC strains of Staphylococcus aureus, Enterococcus faecalis, Escherichia coli and Pseudomonas aeruginosa. The stage-2 solid phase extraction returned 16 fractions, which were tested against S. aureus and E. faecalis. Fractions I, J and K yielded minimum inhibitory concentrations between 16 μg/ml and 256 μg/ml for both Gram-positive.
Despite the presented results, I believe the manuscript findings are preliminary and dose not merit publication. The authors were not able to isolate biologically active compounds and they only identified them using mass spectrometry. Also, the antimicrobial activity was not that potent to support further studies. I think the manuscript cannot be accepted for publication.
Author Response
Dear Reviewer,
Many thanks for reviewing our manuscript entitled “Antibacterial activity and amphidinol profiling of the marine dinoflagellate Amphidinium carterae (subclade III) by Maria Elena Barone – Elliot Murphy, Rachel Parkes, Gerard T.A. Fleming, Floriana Campanile, Olivier P. Thomas and Nicolas Touzet (ijms-1440142).
Please find below our responses to the reviewer’s comments. We hope that the queries have been addressed satisfactorily.
Please do not hesitate to get back to us should further clarifications be needed.
On behalf on the authors,
Yours faithfully,
Maria Elena Barone
#1 Despite the presented results, I believe the manuscript findings are preliminary and dose not merit publication. The authors were not able to isolate biologically active compounds and they only identified them using mass spectrometry. Also, the antimicrobial activity was not that potent to support further studies. I think the manuscript cannot be accepted for publication.
Response: Indeed, no attempt was made to isolate the amphidinol derivatives due to the low amount of biomass available to provide enough material for NMR analyses. However, we are confident about the identification of the known derivatives due to very precise comparison of MS/MS data with comprehensive data published recently by Krock’s team in this family (LC-MS/MS Method Development for the Discovery and Identification of Amphidinols Produced by Amphidinium; Marvin Wellkamp, Francisco García-Camacho, Lorena M. Durán-Riveroll, Jan Tebben, Urban Tillmann and Bernd Krock). Also, we propose that a new derivative might contribute to the detected bioactivity and we therefore believe that our results deserve publication. The next steps will be to identify the structure of the new derivative using more biomass of the cultured microalga.
With regards to the potency of the extracts, antibacterial activity of amphidinol-containing fractions has seldom been reported to date. However, our MIC values against the ATCC reference isolates were in the same range of activity of antibiotics commonly used in the clinical practice (e.g., Chloramphenicol, Fidaxomicin, Cefotetan, Pexiganan, Moxalactam and Ceftazidime) according to the Clinical and Laboratory Standard Institute guidelines (CLSI, 2020).

Reviewer 3 Report
Authors reported antibacterial activity and amphidinol profiling in A. carterae collected in Ireland. The results are important. Therefore, this manuscript is worthy being published in IJMS.
Amphidinols are polyhydroxylated compounds. Is there a possibility that amphidinols are contained in the H2O layer after the partition EtOAc/H2O?
Author Response
Dear Reviewer,
Many thanks for reviewing our manuscript entitled “Antibacterial activity and amphidinol profiling of the marine dinoflagellate Amphidinium carterae (subclade III) by Maria Elena Barone – Elliot Murphy, Rachel Parkes, Gerard T.A. Fleming, Floriana Campanile, Olivier P. Thomas and Nicolas Touzet (ijms-1440142).
Please find below our responses to the reviewer’s comments. We hope that the queries have been addressed satisfactorily.
Please do not hesitate to get back to us should further clarifications be needed.
On behalf on the authors,
Yours faithfully,
Maria Elena Barone
#1 Amphidinols are polyhydroxylated compounds. Is there a possibility that amphidinols are contained in the H2O layer after the partition EtOAc/H2O?
Response: amphidinols are well studied derivatives and are amphiphilic due to the presence of an alkyl chain on the lower side of the molecule. As other researchers working on these compounds, we observed that they have a very low solubility in water but are highly soluble in methanol. We therefore believe that only a small amount of these derivatives are lost in the water fraction.

Round 2
Reviewer 2 Report
International Journal of Molecular Sciences (Manuscript ID: ijms-1440142), Comments to the Authors:
Title: Antibacterial activity and amphidinol profiling of the marine dinoflagellate Amphidinium carterae (subclade III)
Comments:
After reading the authors response to my comments, I believe the authors did not any new information to respond to my comments. I believe that the results are preliminary and further in-depth analysis is needed to draw a sloid conclusion of the authors’ results. The authors mentioned the following sentence, “The next steps will be to identify the structure of the new derivative using more biomass of the cultured microalga.” I believe after they identify the new derivatives and determine their MIC the results may be considered for publication.